# Investigating the Effect of Operating Parameters on the Wear of Abrasive Tools in the Polishing Stage of Granitic Building Stones

Ali Farhadian, Ebrahim Ghasemi *, Seyed Hadi Hoseinie and Raheb Bagherpour

Department of Mining Engineering, Isfahan University of Technology, Isfahan 8415683111, Iran
* Correspondence: e_ghasemi@iut.ac.ir

**Abstract:** Operating parameters affect the wear of abrasive tools during the polishing stage in building stone processing plants. This study investigates the effects of essential operating parameters including polishing head pressure, head rotation speed and water flow rate on the wear of the abrasive tools. For this purpose, a building stone abrasivity test was used to determine the weight loss of the abrasive tools during laboratory polishing of fifteen different types of Iranian granitic building stones. The standard operating parameters of the test were a polishing head pressure of 5 bar, a head rotation speed of 300 revolutions per minute (rpm), and a water flow rate of 4 L/min. The values of the operating parameters were changed to values within the range from ±25% and ±50% of the standard conditions in order to investigate the effect of variations in these parameters on the wear of the abrasive tools during the polishing stage. The results of different tests showed that the wear of the abrasive tools was directly proportional to the pressure up until a critical value of around 6.25 bar, after which it gradually decreased. This nonlinear wear behavior does not conform to Archard's well-known classical wear law. The FESEM images of the worn surfaces showed that due to excessive load, debonded abrasive particles could not be pulled out from the pin surface and led to an interlocking phenomenon between the pin and stone surface. It was also found that the wear of the abrasive tools increased with increasing head rotation speed, while it decreased with the water flow rate. Moreover, the main wear mechanism of tests was abrasive wear and in some cases with a mixture mode of adhesion and delamination.

**Keywords:** wear; granitic building stones; abrasive tools; polishing process; operating parameters



## 1. Introduction

In the building stone industry, there are several methods for stone surface finishing, including polishing, honing, flaming, tumbling, brushing, etc. One of the best methods for increasing the surface quality of stone is the polishing operation [1].

Nowadays, stone polishing operations are performed using long polishing machines consisting of successive heads (Figure 1). Six abrasives are radially located in a head and passed against the stone tile surface. One of the main concerns of manufacturers regarding the polishing process is the wear and removal of the abrasive tools. According to previous studies, more than 40% of the product's total costs are related to the polishing stage [2–5]. Many of these costs are related to the high rate of consumption of abrasive tools, because about half a kilogram is required per square meter of the final product [6]. Hence, it is imperative to identify the predominant wear mechanism and investigate the effective parameters on the wear of the abrasive tools.

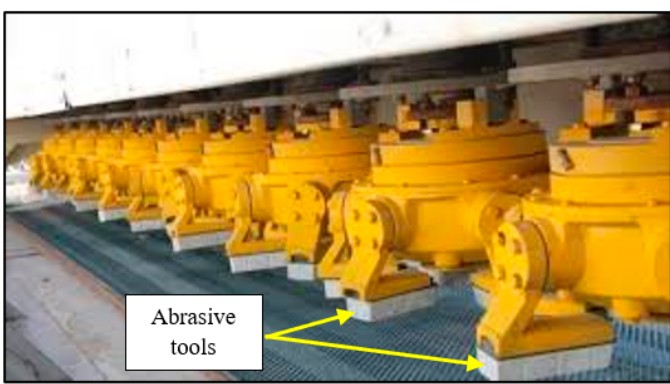

**Figure 1.** A long polishing machine with successive heads.

Wear is the removal of material from the material surface due to the mechanical operations of bodies on each other, and the wear mechanism refers to the physical and chemical processes occurring during wear [7]. Many classifications of wear types have been presented in the literature, one of the most commonly of which is based on the wear mechanism, such as abrasive wear, adhesive wear, fatigue wear, and delamination wear. The details of such wear types can be found in the literature [7–14]. Based on previous estimations, abrasive wear accounts for about 50% of industry wear [15,16]. According to the kind of contact, abrasive wear is classified into two-body and three-body abrasive wear. In two-body wear, hard asperities without separating generate many grooves on the other surface (like sandpaper), but in three-body wear, abrasive grains are located between two soft surfaces and separate material from them [9]. Besides the type of deformation, the properties of the solid and counter body, the interfacial element and the loading conditions determine the wear mechanisms [11]. The volume loss due to abrasive wear is calculated as follows (Archard's equation):

$$V = (K.L.F)/H \tag{1}$$

where V is the volume loss ($m^3$), K is the wear coefficient, L is the sliding distance (m), F is the applied load (N) and H, is material hardness (Pa) [11,17,18]. This equation can be used to influence the operating parameters of the abrasive wear of a wide range of materials [8,19].

It should be noted that abrasivity is the main factor in removing the abrasive tools during the polishing process. The abrasivity is not an inherent attribute of materials, and depends on operating parameters, material properties, and tribo-system characteristics [13]. In other words, the wear of materials is characterized by the fact that each tribo-system is unique. Therefore, there will be several wear mechanisms under different conditions [20].

In the polishing process of building stones, there are some factors affecting the wear of the abrasive tools, including the physico-mechanical and petrographic properties of stone, the composition of the abrasive tool, and operating parameters. Operating parameters include polishing head pressure, head rotation speed, and water flow rate. The effect of the physico-mechanical and petrographic properties of the granitic stone on abrasive tool wear were studied in our previous work [20], and the composition of the abrasive tool is beyond the scope of the present study. However, based on the authors' field observations in various Iranian processing plants of building stones, the operating parameters are the most crucial factors in the wear of abrasive tools. It is noted that one of the most important economic parameters in the granitic polishing stage is the loss of abrasive tools. Therefore, it is essential to study the effect of operating parameters on the wear of abrasive tools.

A review of the literature reveals that numerous studies have investigated building stones in the polishing stage. However, they were focused mainly on optimizing operating parameters to address aesthetic issues (roughness and glossiness), rather than considering the material removal of the abrasive tools and their effective mechanism. Barbosa et al. changed the polishing head pressure and water flow rate to achieve maximum glossi-

ness of Portuguese limestone in processing plants. They found that the best operating parameters for these kinds of stones were a head pressure of 2 bar and a water flow rate of 30 L/min [1]. Yavuz et al. investigated the effect of conveyor belt speed on the surface quality of Turkish carbonate stones. They carried out polishing tests via an automatic belt polishing machine specially designed for laboratory experiments by maintaining the head pressure, head rotation speed and water at constant values. They concluded that increased belt speed causes enhanced roughness of the stone's surface and decreased glossiness [4]. Ersoy et al. investigated the effect of abrasive head rotation on surface quality and revealed that smoother and brighter surfaces could be obtained by increasing the abrasive head's rotational speed [21]. However, regarding the polishing process of building stone, some studies have also been carried out regarding the effect of physico-mechanical properties of granitic and carbonate building stones, as well as the function of abrasive tool composition on stone surface quality. Cevheroglu et al. investigated the effects of the material properties of four limestones on surface roughness and glossiness under fixed operating conditions. Their research showed that surface roughness increased with increasing stone porosity, whereas it decreased with increasing uniaxial compressive strength [22]. Gorgulu and Ceylanoglu investigated the effects of diamond and SiC abrasives on surface quality and discovered that the surface roughness and glossiness of the stone samples they examined were independent of the abrasive type used [23]. Additionally, there have been some other publications investigating the effect of stones' physico-mechanical properties and abrasive tool composition on the surface quality of building stones [24–29]. As mentioned earlier, the previous literature does not provide detailed quantitative information on the wear of abrasive tools during the polishing process. Therefore, the wear behavior and mechanism of abrasive tools in the polishing process are not yet well understood.

In 2021, a new laboratory-scale abrasivity test was developed by the present authors [20]. However, the effect of polishing operating parameters on the wear of abrasive tools wasn't investigated. Therefore, in this work, the main objective is a comprehensive experimental study to examine the influence of the operating parameters on the abrasive tool wear. Additionally, the polished surfaces of the abrasive tools are characterized in terms of their mechanisms and scrutinized using field emission scanning electron microscopy (FESEM).

## 2. Materials and Methods

### 2.1. Materials

To perform this study, fifteen samples of Iranian commercial granites were prepared from various stone processing plants. The samples have different mineralogical compositions, and are free from any visible cracks or indications of weathering (Figure 2). To reduce the life of the abrasive tools during the tests, granitic samples with high hardness and abrasiveness were selected.

It should be noted that the term "granite" has two different definitions: scientific and commercial. Granite is scientifically defined as a crystalline and hard igneous stone composed of quartz, feldspars, and accessory minerals such as mica. In contrast, commercial granite covers all hard and crystalline igneous stones with different petrographic properties that can be polished well [19]. For all stone types, block samples with sufficiently large dimensions were provided by stone processing plants and brought to the laboratory for sampling and testing.

To identify the studied samples, their quantitative petrographic and physico-mechanical properties were determined. For petrographic characteristics, two thin sections, parallel and perpendicular to the sample axis, were prepared from each stone type. For preparation of the thin section, a suitable size slab was first cut with a diamond saw; after that, the slab was labeled on one side and the other side was lapped flat and smooth, first on a cast iron lap with 400 grit carborundum, then by polishing on a glass plate with 600 grit carborundum. In the next step, after drying on a hot plate, a glass slide was glued to the lapped face of the slab with epoxy. Finally, using a thin section saw, the slab was cut off

close to the slide. After providing the thin section, under a polarized microscope (Nikon Eclipse LV100POL), different percentages of minerals were detected, thus determining the modal composition of the studied samples. At the end of the determination of the minerals, the stone samples were classified based on the Streckeisen classification system [30]. The results of the petrographic studies are given in Table 1.

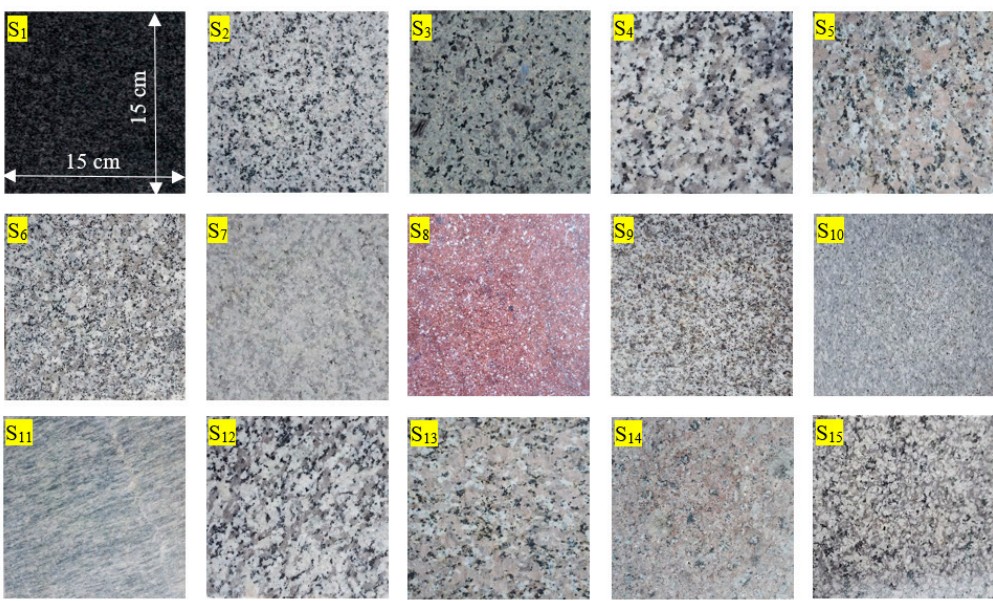

**Figure 2.** Different granitic building stones used in this study.

**Table 1.** The results of the petrographic studies of the samples.

| Sample | Commercial Name | Scientific Name * | Modal Composition (%) | | | | | | | | | | |
|--------|-----------------|-------------------|------|------|------|------|------|------|------|------|------|------|------|
| | | | Qz | Pl | Or | Bt | Mu | Amp | Pr | Ch | Op | Tu | Ma |
| S₁ | Meshki-Natanz | Diorite | 10 | 60 | - | 5 | - | 15 | 2.5 | - | 7.5 | - | - |
| S₂ | Sefid-Natanz | Granodiorite | 23 | 51 | 5 | 10 | - | 8 | - | - | 3 | - | - |
| S₃ | Khorramdarreh | Syenogranite | 20 | 15 | 50 | 8 | - | - | - | - | 2 | 5 | - |
| S₄ | Golpanbeh-Nehbandan | Granite | 28 | 26 | 35 | 11 | - | - | - | - | - | - | - |
| S₅ | Tiybad | Syenogranite | 30 | 20 | 45 | 5 | - | - | - | - | - | - | - |
| S₆ | Borujerd | Granite | 22 | 38 | 30 | 8 | - | - | - | 2 | - | - | - |
| S₇ | Zahedan | Granite | 47.5 | 23 | 19 | 3 | 7.5 | - | - | - | - | - | - |
| S₈ | Ghermeze-Yazd | Andesite | 20 | 15 | - | 3 | - | - | - | - | 2 | - | 60 |
| S₉ | Morvarid-Mashhad | Granite | 23.5 | 19 | 43.5 | 8.5 | 5.5 | - | - | - | - | - | - |
| S₁₀ | Shaghaegh-Nehbandan | Monzonite | 19 | 33.5 | 36 | 9.5 | - | - | - | - | 2 | - | - |
| S₁₁ | Sabze-Birjand | Granite | 32 | 38 | 15 | 5 | 15 | - | - | - | - | - | - |
| S₁₂ | Tucy-Astan | Granite | 40 | 20 | 28 | 5 | 7 | - | - | - | - | - | - |
| S₁₃ | Porteghly-Nebandan | Granite | 40 | 25 | 30 | 3 | - | - | - | 2 | - | - | - |
| S₁₄ | Holoee-Zanjan | Syenite | 8 | 8 | 75 | - | - | 6 | 3 | - | - | - | - |
| S₁₅ | Maraghe | Syenogranite | 25 | 16 | 54 | 3 | 2 | - | - | - | - | - | - |

\* According to the Streckeisen classification system. **Note:** Qz: Quartz, Pl: Plagioclase, Or: Orthoclase, Bt: Biotite, Mu: Muscovite, Amp: Amphibole. Pr: pyroxene, Ch: Chlorite, Op: Opaque, Tu: Tourmaline, Ma: Matrix (iron dioxide).

It should be noted that our previous paper presented the physico-mechanical properties of the studied samples, namely, apparent density, effective porosity, uniaxial compressive strength, and Brazilian (indirect) tensile strength [20].

### 2.2. Laboratory Abrasivity Tests

All of the abrasivity tests were performed in the laboratory, using the new abrasivity rig described in our previous publication [20]. Due to the novelty of the test, the new rig and the test method are explained briefly. The rig includes a working table, a polishing head, a pressure control unit, an electrical driving force unit (electro-gearbox), a speed control unit, and a water flow regulation unit (Figure 3).

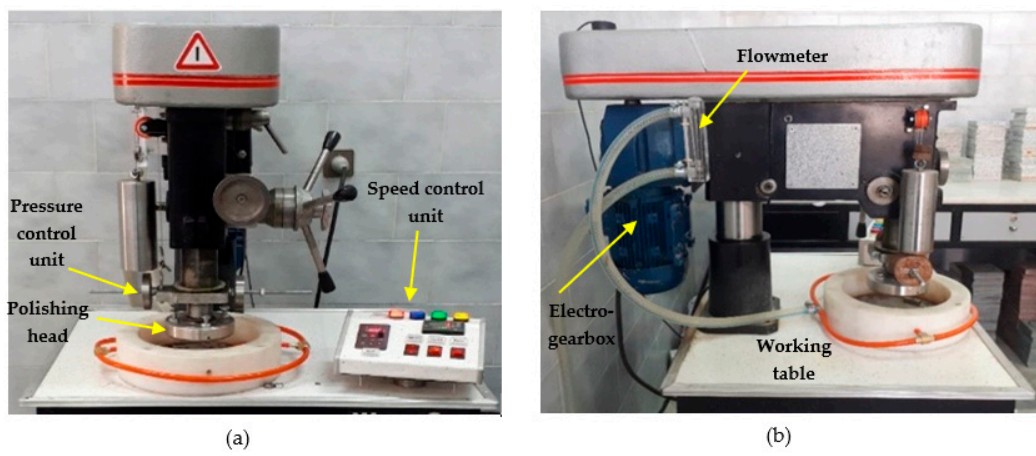

**Figure 3.** Laboratory test rig: (**a**) front view; (**b**) side view.

Cylindrical abrasive pins 8 mm in diameter and 30 mm in length were prepared from the abrasive block employed in the industry with the grit number 120, using water jet cutting technology (Figure 4). Based on the XRD results, the abrasive tool consists of silicon carbide (SiC) particles (5%), as abrasive material, bonded together by a magnesium resin (49%), as cement, calcium oxide (7%) and loss-on-ignition materials (39%). These results were obtained using an X-Ray Diffraction spectrometer (model: Shimadzu XRD-6100), by means of the parallel beam method with a scan angle between 10 and 100 degrees.

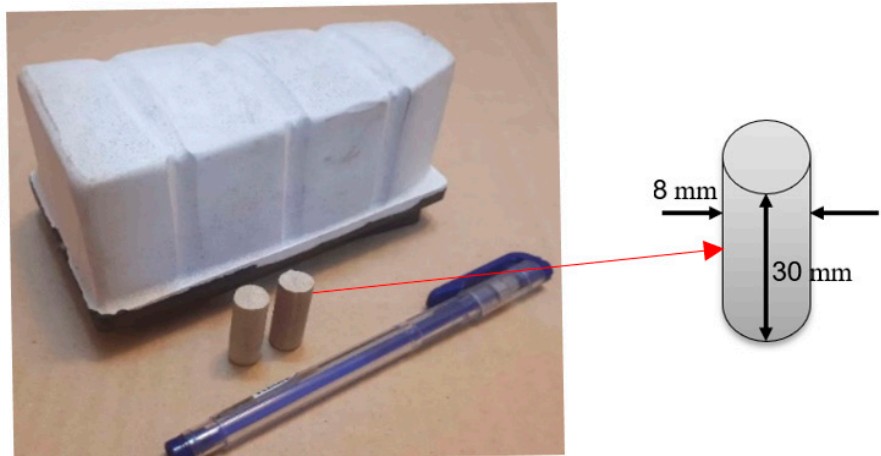

**Figure 4.** Industrial abrasive block and extracted abrasive pins.

Two abrasive pins were used for each test and mounted in the rotating head. The experiment was performed on a square stone tile with a width of 15 cm and a thickness of 2 cm. It was located in the Teflon chamber and fixed by holders. Moreover, the stone

tile surface should be planar and smooth (for this purpose, calibrated stone tiles with diamond abrasives could be used). Figure 5 shows a schematic diagram of the relative motion between the abrasive pin and the stone sample.

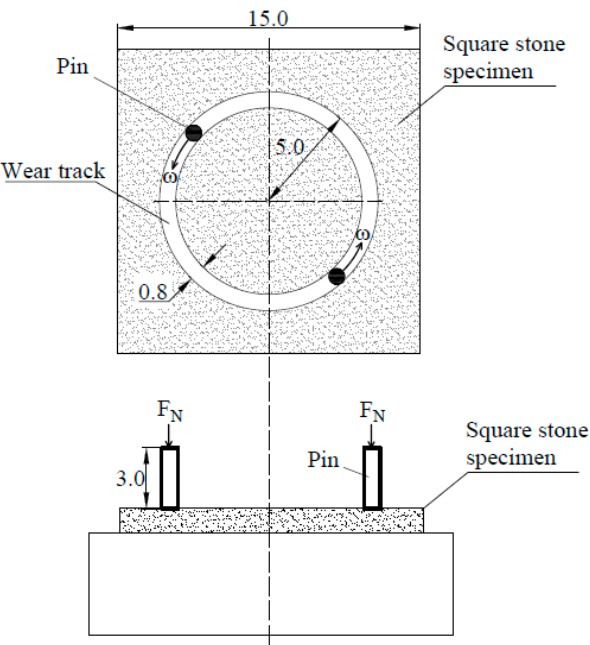

**Figure 5.** Schematic diagram of movement of the pins on the stone sample in the new rig ($\omega$ is the rotation speed).

The pre-weighed abrasive pins were mounted in the motorized polishing head with their axes vertical to the plane of the stone specimen as shown in Figure 5; they rotated about a vertical axis in the head of the rig with the rotational speed of 300 revolutions per minute (rpm). The pins were pressed against the stone specimen surface under a normal pressure of 5 bar. It should be noted that according to the dimension of pins used in the test, the sum of the areas of the two pins was equal to one square centimeter. As a result, the values of pressure and load is the same. Moreover, a water flow rate of 4 L/min was considered for flushing and cooling purposes. The values were nearly in accordance with the real working conditions during the building stones' polishing process, and can therefore be considered standard operating conditions. As recommended in [20], the testing time duration was five minutes. Once the test was finished, the amount of abrasive tool wear, i.e., the weight loss of the two miniature pins, was measured in grams, as follows:

$$\text{Abrasive tool wear} = M_0 - M_1 \tag{2}$$

where $M_0$ and $M_1$ are the weight of the abrasive pins before and after the test, respectively, in grams.

In the designed experiments, to study the effect of operating parameters on the wear of abrasive tools during the polishing process, the standard operating parameters were changed in intervals between ±25% and ±50%. Accordingly, the abrasivity tests were carried out on fifteen different types of Iranian granitic building stones by applying various operating parameters, including polishing head pressure (2.5, 3.75, 6.25 and 7.5 bar), polishing head rotation speed (150, 225, 375 and 450 rpm), and water flow rate (2, 3, 5 and 6 L/min). Moreover, to better show the behavior of abrasive tool wear, a further study, including the water flow rate of 7 L/min, was also conducted for all studied samples. Therefore, besides the standard abrasivity tests, 13 other test modes were employed for each stone sample by applying the above values of the operating parameters. The details of all abrasivity tests are presented in Table 2. Finally, to eliminate the uncertainty of the experiment results, three tests were considered for each stone type (in total, 630 tests).

**Table 2.** Operating parameters values for the abrasivity tests.

| Operating Parameters | Test Modes | Variation Percent | Head Pressure (Bar) | Head Rotation Speed (rpm) | Water Flow Rate (L/min) |
|---|---|---|---|---|---|
| Head pressure | 1 | −0.50% | 2.5 | 300 | 4 |
| | 2 | −0.25% | 3.75 | 300 | 4 |
| | 3 | base | 5 | 300 | 4 |
| | 4 | +0.25% | 6.25 | 300 | 4 |
| | 5 | +0.50% | 7.5 | 300 | 4 |
| Head rotation speed | 6 | −0.50% | 5 | 150 | 4 |
| | 7 | −0.25% | 5 | 225 | 4 |
| | 8 | +0.25% | 5 | 375 | 4 |
| | 9 | +0.50% | 5 | 450 | 4 |
| Water flow rate | 10 | −0.50% | 5 | 300 | 2 |
| | 11 | −0.25% | 5 | 300 | 3 |
| | 12 | +0.25% | 5 | 300 | 5 |
| | 13 | +0.50% | 5 | 300 | 6 |
| | 14 | +0.75% | 5 | 300 | 7 |

## 3. Results and Discussion

In this study, an average of at least three measurements is reported in Table 3 based on standard conditions. It should be noted that the effect of environmental temperature on the test was assumed to be negligible, because all tests were carried out at ambient temperature, i.e., 25 °C. As can be seen in Table 3, samples 7 and 13 ($S_7$, $S_{13}$) have higher abrasivity than the other samples. Additionally, samples 1 and 11 ($S_1$, $S_{11}$) have the lowest abrasivity, while the other samples belong to the medium abrasivity class. It should be noted that the presence of abrasive minerals such as quartz causes an increase in stone abrasivity, while cleavable minerals, including plagioclase and micas (biotite and muscovite), have cleavage planes that result in decreased stone abrasivity [20]. As reported in Table 1, the samples with high abrasivity ($S_7$, $S_{13}$) have 47.5 and 40 percent quartz, respectively. Additionally, these samples have a low percentage of cleavable minerals. On the other hand, samples 1 and 11 ($S_1$, $S_{11}$) have 65 and 58 percent cleavable minerals.

**Table 3.** Results of building stone abrasivity tests in the standard conditions.

| Sample | Abrasive Tool Wear (g) |
|---|---|
| $S_1$ | 0.244 (±0.021) |
| $S_2$ | 0.320 (±0.016) |
| $S_3$ | 0.351 (±0.021) |
| $S_4$ | 0.444 (±0.020) |
| $S_5$ | 0.523 (±0.044) |
| $S_6$ | 0.325 (±0.023) |
| $S_7$ | 0.717 (±0.045) |
| $S_8$ | 0.301 (±0.005) |

**Table 3.** *Cont.*

| Sample | Abrasive Tool Wear (g) |
|:---:|:---:|
| $S_9$ | 0.403 ($\pm$0.025) |
| $S_{10}$ | 0.586 ($\pm$0.020) |
| $S_{11}$ | 0.254 ($\pm$0.029) |
| $S_{12}$ | 0.650 ($\pm$0.082) |
| $S_{13}$ | 0.705 ($\pm$0.015) |
| $S_{14}$ | 0.429 ($\pm$0.028) |
| $S_{15}$ | 0.469 ($\pm$0.032) |

Values given in parentheses represent standard deviation.

### 3.1. The Effect of Polishing Head Pressure on Abrasive Tool Wear

To understand the behavior of the wear phenomenon, laboratory abrasivity tests were carried out while changing the normal contact pressure. As presented in Table 2, by maintaining the head rotation speed at 300 rpm and the water flow rate at 4 L/min, five levels of pressure were considered while performing the abrasivity tests: 2.5, 3.75, 5, 6.25 and 7.5 bar. Figure 6 presents the test results of the studied samples when subjected to various pressures. According to Figure 6, abrasive tool wear first increased, reaching a peak at a critical pressure of around 6.25 bar, after which it gradually decreased. This behavior is inconsistent with the Archard's classical wear law (Equation (1)). Based on this law, the wear rate should increase with increasing load. Although many researchers have shown that the amount of wear also increases with increasing pressure in various materials [15,31–33], there have been some studies illustrating that the amount of wear decreases with increasing pressure [34,35].

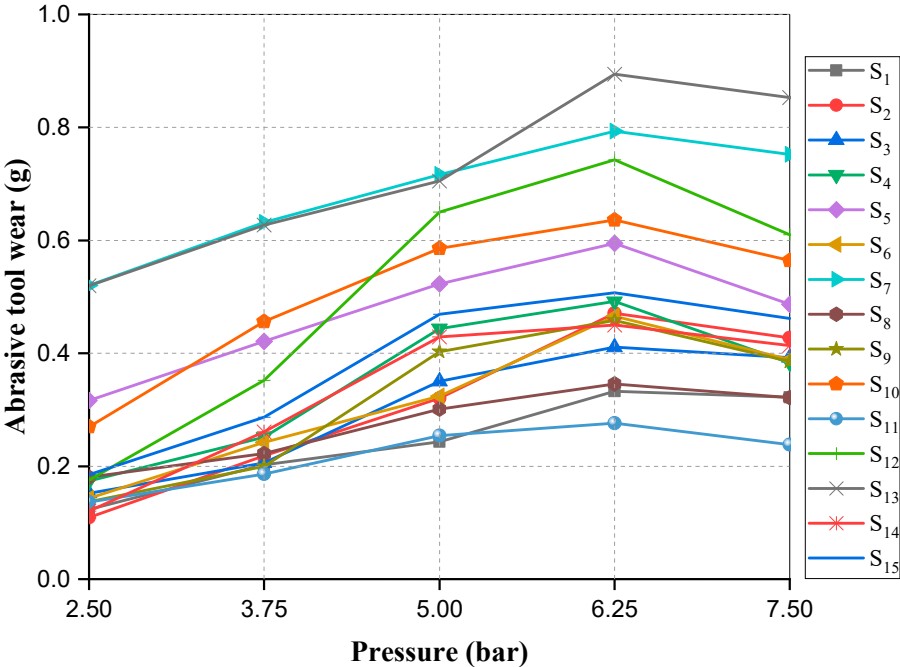

**Figure 6.** Variation in wear of the abrasive tools with applied pressure.

In the abrasivity test, both the pin and stone surfaces have many asperities, differing in shape and height. It could be said that there is no perfect smooth sliding surface. The granitic stone surface was made up of hard minerals such as quartz and feldspars that were harder than the pin surface. According to the measurements, the hardness of the studied stones were several times higher than that of the abrasive tool (samples Vickers

hardness: 495–815 HV and abrasive tool harness: 50 HV). It should be noted that abrasive wear can occur at a low or high level depending on the ratio of the abrasive hardness to the hardness of the surface being worn (hardness contrast). The contrast in the hardness of the two solid surfaces is a fundamental factor in abrasive mechanisms. Contrasts in hardness greater than two or less than 0.7 indicate the high and low levels of the abrasive wear, respectively [11]. Therefore, the high-level abrasive wear prevails in the abrasivity test. It is needless to say that the SiC particles of the pin are harder than the stone minerals (2690 HV); however, based on the XRD results, their percent volume was low (5 Wt.%). Hence, the harder asperities of the stone surface began to cut the surface of the pin penetrating its subsurface (Figure 7). In this case, the applied load determined the degree of penetration of the stone asperities into the pin surface. The penetration increased with the increase in applied pressure, causing an increase in pin removal. Moreover, the applied pressure dictates the intimacy of the contact surface. The increase in applied pressure leads to a more intimate contact between the pin and the stone surface. This caused further friction and adhesion, resulting in the softening of the pin surface. Accordingly, the penetration of stone surface asperities into the pin surface became easier, leading to the removal of more material up until the critical pressure. After the test, the microscopic grooves on the pins confirmed that the abrasive wear process was the predominant wear mechanism.

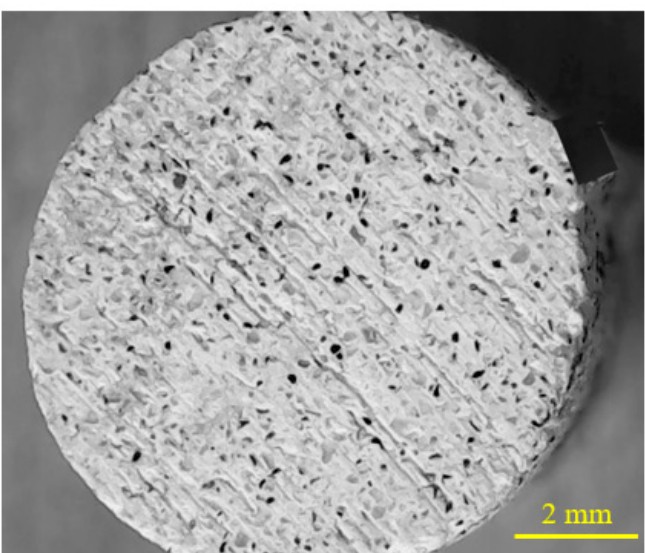

**Figure 7.** Worn pin surface after the test; abrasive wear due to the penetration of stone asperities into the pin surface.

Wear mechanisms are highly dependent on changes in pressure and stress created at the surfaces in contact. At low pressure, there was an elastic contact between the sample surfaces and abrasive materials. With increasing load, there was an elastic–plastic contact transition on the contact surfaces, and the contact area was insensitive to the loads; thus, this led to a stable relationship between the friction coefficient and the applied loads, as well as a stable relationship between the wear rates and the applied loads [35]. The high local pressure between the contacting asperities resulted in plastic deformation and, adhesion, consequently leading to the local formation of junctions. Relative sliding between the contacting surfaces caused the rupture of these junctions, frequently transferring material from one surface to the other [11].

The appearance of the worn surfaces in a tribo-system can indicate which wear mechanism was acting [12]. Therefore, the reasons for the nonlinear behavior were inspected through field emission scanning electron microscopy (FESEM) micrographs. In this study, all micrographs were produced using an FEI microscope set (model QUANTA FEG-450) under high-vacuum conditions and at a voltage of 25,000 volts. Figure 8 presents the

micrographs of the worn surfaces of the pin as a function of the applied pressure. As can be seen, due to many cuts and ploughs, the abrasive wear mechanism predominated. It should be noted that wear is usually caused by different mechanisms; however, only one is commonly considered to be the main mechanism or the controller of the process [36].

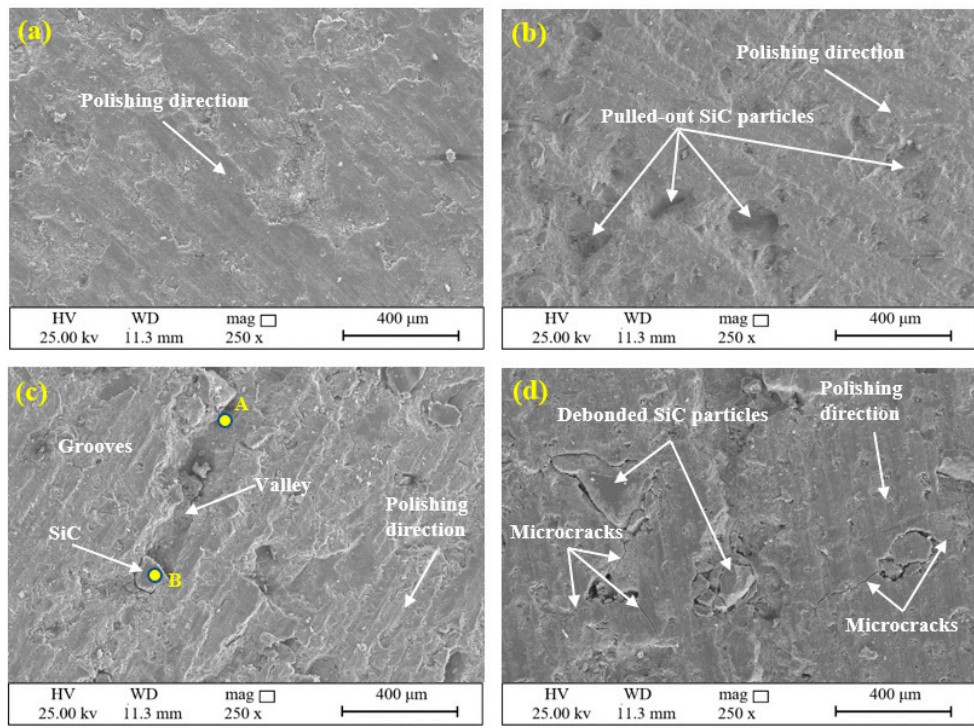

**Figure 8.** FESEM micrographs of the worn pin surfaces after performing the test on granodiorite stone (S$_2$) with different pressures: (**a**) 2, (**b**) 3.75, (**c**) 6.25 (A and B are sliding points), and (**d**) 7.5 bar.

According to Figure 8a, at a pressure of 2.5 bar, the worn surface appeared relatively smooth and flat. Micro-ploughing and micro-cutting caused by the hard abrasive minerals could be seen on it. There were no deep grooves, and as a result, the subsurface remained intact. Moreover, the contribution of adhesive wear and delamination seems to be negligible; therefore, the amount of overall wear was low. Additionally, in this case, the dominant wear mechanism was abrasive wear. With increasing values of pressure, the friction and the plastic deformation on the worn surface increased, leading to the formation of some micro pits due to the detachment of SiC particles (Figure 8b). As can be seen in this figure, in addition to abrasive wear, adhesive wear and delamination, which had a greater impact than in the previous case, the formation of pits also had a further effect on the weight loss of the pin. Therefore, the amount of wear also increased. At a pressure of around 6.25 bar, the combination of abrasive, adhesive and delamination mechanisms considerably affected the wear loss of the pin. In this case, the subsurface of the pin was affected, and many deep grooves were created on the pin surface, thus further confirming the penetration of stone surface asperities. Sometimes, detached SiC particles from the pin became trapped between the contacting surfaces (three-body abrasive wear) before leaving the environment, sliding from point A to point B. As a result, valleys were created on the surface of the pin (Figure 8c). Finally, due to excessive load at a higher pressure of around 6.25 bar, the wear mechanisms were not completely formed. The debonded abrasive particles were not able to pull out from the pin surface, i.e., interlocking between the pin and the stone surface had occurred (Figure 8d), consequently decreasing the tool wear. Another noteworthy point in Figure 8d is the presence of microcracks that started and grew from the concentration points of stress. It is possible that, with increasing pressure, the microcracks will also increase, and their joining together will lead to the rupture of the pin. However, in polishing plants

for building stones, pressures greater than 7.5 bar are rarely used. Therefore, in this study, no pressures greater than this were applied.

### 3.2. Effect of Head Rotation Speed on Abrasive Tool Wear

The sliding distance was increased by increasing the rotation speed. According to Equation (1), when the sliding distance on a stone surface is increased, the amount of wear volume increases, too. In this section, to investigate the effect of the head rotation speed on the abrasive tool wear, five speeds—150, 225, 300, 375 and, 450 rpm—were applied (Table 2). The test was repeated at least three times for all studied stones. The average results are plotted in Figure 9. This figure provides the details of the abrasive tool wear when varying the head rotation speed (sliding distance) at a constant head pressure of 5 bar and a fixed water flow rate of 4 L/min. As can be seen from Figure 9, the abrasive tool wear increased with increasing head rotation speed. The increase for samples with high abrasivity ($S_7$, $S_{13}$) was slightly smaller than in the other samples. Additionally, for the other samples, the abrasive tool wear was approximately proportional to head rotation speed (sliding distance). This behavior is in accordance with Archard's law (Equation (1)).

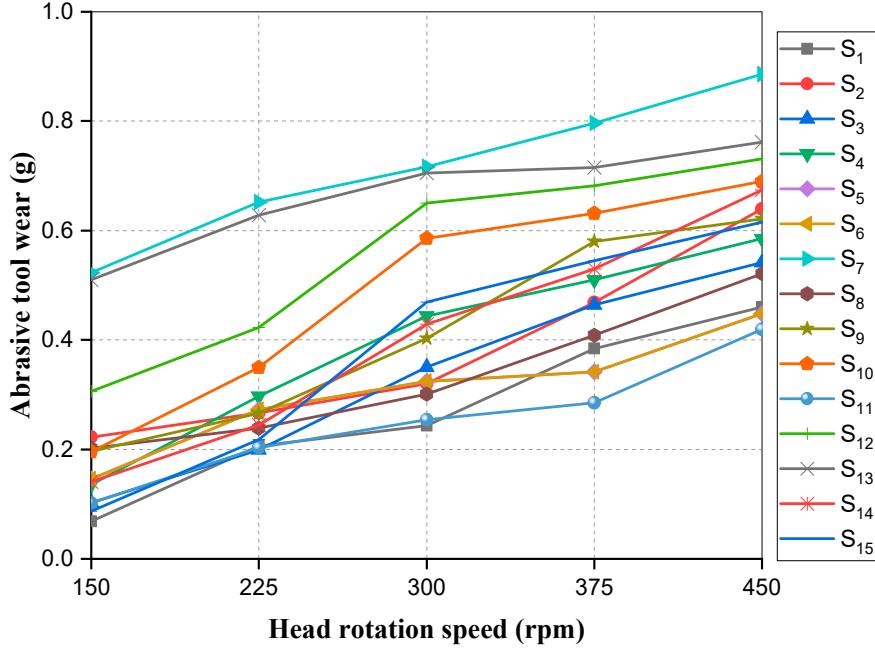

**Figure 9.** Variation in abrasive tool wear with head rotation speed.

To further analyze this behavior, FESEM photomicrographs were prepared for all samples. Figure 10 shows the FESEM images of the worn pin after testing on granodiorite stone ($S_2$). According to Figure 10a, abrasive wear was the dominant mechanism at low rotation speeds. With increasing speed, the plastic deformation and transition area from abrasive to adhesive wear increased, causing the formation of many micro pits and micro cutting on the worn area (Figure 10b). As can be seen from the figure, a series of influencing factors caused the amount of wear to be greater than in the previous case.

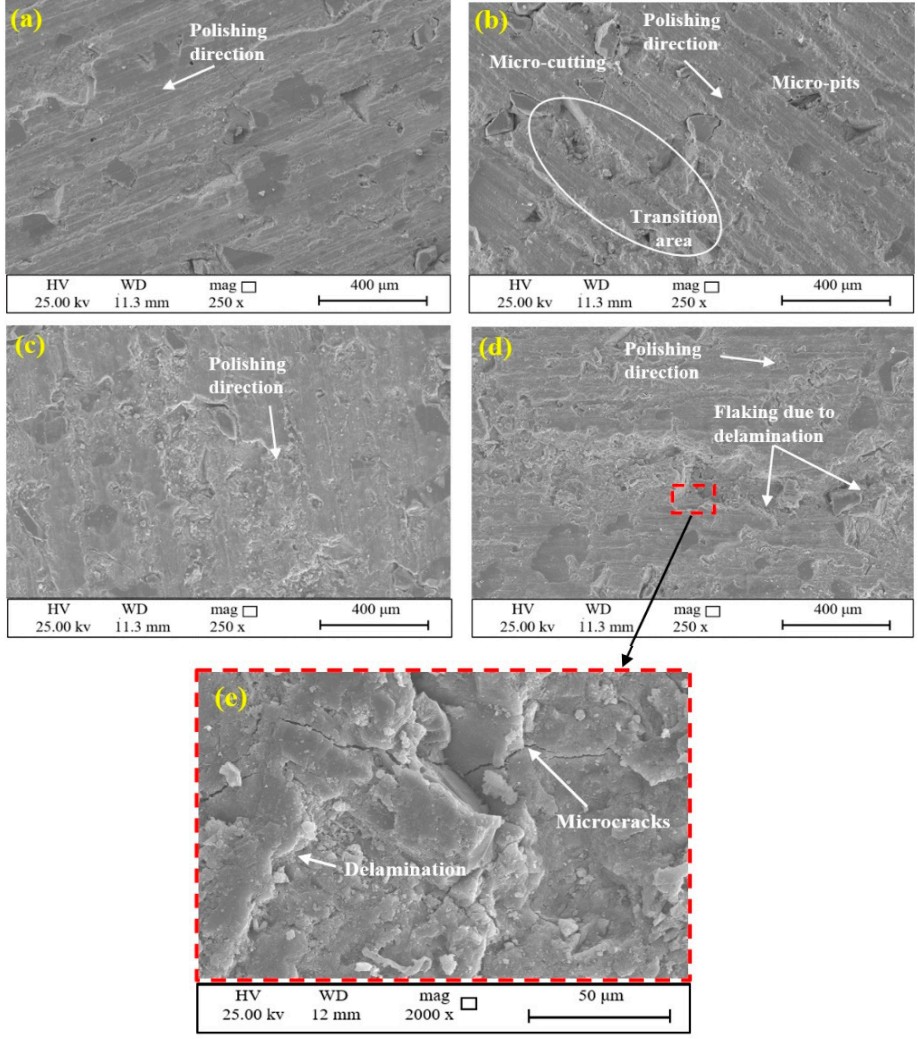

**Figure 10.** FESEM micrographs of the worn pin surfaces after performing the test on granodiorite stone ($S_2$) with different head rotation speeds: (**a**) 150, (**b**) 225, (**c**) 375,(**d**) 450 rpm and (**e**) a close- up of delamination wear.

At high rotation speeds (450 rpm), flakes of delamination can be clearly seen on the surface of the pin (Figure 10d,e), which may be because of the development of instability in the tribo-layer at longer sliding distances. In other words, the worn pin surface disintegrated as a result of the simultaneous action of adhesion, abrasion and delamination wear mechanisms. Therefore, there was an increase in the amount of abrasive tool wear. These results are in agreement with the findings of researchers such as Odabas [15], Zhang et al. [35], and Kumar et al. [37]. Although the samples they used were ceramic tile and aluminum, the trends they reported for abrasive tool wear were similar to those reported above.

### 3.3. Effect of Water Flow Rate on Abrasive Tool Wear

The surface area coming into actual contact has to be considered a heat source acting over only a short time. The temperature distribution of the surfaces in contact is strongly dependent on surface pressure, velocity, lubricant, contact geometry, conductivity, etc. [11].

In the polishing process of building stones, the contacting surfaces between the abrasive tools and stone tile are flooded with water, in order to remove heat and flush away the wear debris from the area. Therefore, to evaluate the effect of the water flow rate on the abrasive tool wear, the experiments were carried out by considering water flow rates of 2, 3, 4, 5, 6 and 7 (L/min) with a constant head rotation speed of 300 rpm and a fixed

pressure of 5 bar. The results obtained for these tests are illustrated in Figure 11. As can be seen, when the water flow rate was increased to 5 L/min, the wear of abrasive tool decreased slightly, after which it dropped suddenly for all almost samples. This behavior is probably due to the presence of an adequate water film, as a cooling factor, at pressures between 5 and 6 L/min, causing a reduction in friction and temperature in the contact area. Moreover, in the studied samples, the wear of the abrasive tool was nearly constant at water flow rates greater than 6 L/min.

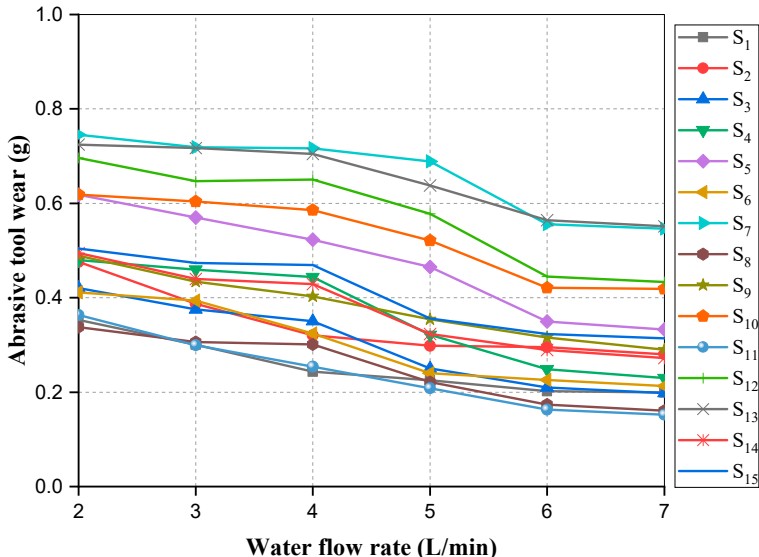

**Figure 11.** Variation in abrasive tool wear with the water flow rate.

According to the observations performed during the tests at a low flow rate (2 L/min), the water was not able to completely flush away the wear debris (Figure 12). To better understand the reasons for the high weight loss at a flow rate of 2 L/min, the temperatures of the wear track were measured immediately after the test using a thermometer device (brand CEM, model: DT-8861). As shown in Figure 12, the temperature of the wear track increased to about 52 °C immediately after the test. Due to the inaccessibility of the wear track during the tests, the temperature was recorded after the test, i.e., during the cooling time of the wear track. It was obvious that the real temperature of the wear track during the test was greater than this value. Therefore, increasing the temperature and friction increases the plastic deformation at the pin surface, subsequently increasing the abrasive tool wear.

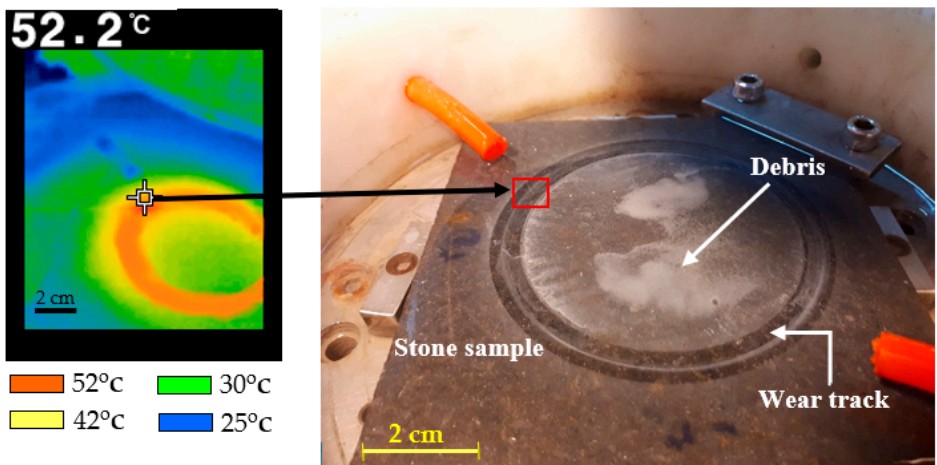

**Figure 12.** Performing the test at a flow rate of 2 L/min and weak flushing of debris.

The FESEM micrographs of the worn pin surfaces at a low flow rate showed that there was insufficient water located between the pin and the stone surface; therefore, friction and interaction were increased. This resulted in an increase in the temperature of the tribo-system; there could also be in increase in the plastic deformations of the pin surface and its subsurface. According to Figure 13a,b, abrasive and adhesive wear can be regarded as the main mechanisms in the weight loss of the abrasive tools. However, due to high friction and adhesion, microcracks grew, and the silicon carbide particles of the pin were debonded from the surface earlier than scheduled. Therefore, there was an increase in the abrasive tool wear. In contrast, water was able to play its role properly at high flow rates, i.e., it was able to reduce friction and SiC particle debonding. With increasing water flow rate (Figure 13c,d), the adhesion wear mechanism transitioned to abrasive wear, i.e., the main mechanism was abrasive wear only. Therefore, the amount of abrasive tool wear decreased by 7 L/min, and remained nearly constant.

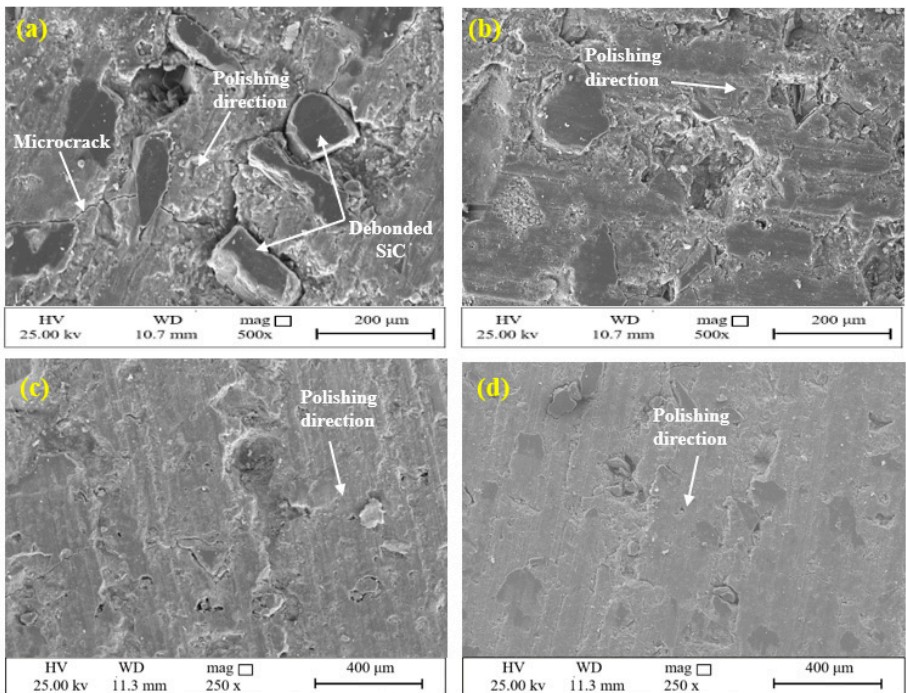

**Figure 13.** FESEM micrographs of the worn pin surfaces after performing the test with different water flow rates: (**a**) 2, (**b**) 3, (**c**) 6 and (**d**) 7 L/min.

Generally, in multiparameter tests, one of the factors will have the highest impact on the test results. To evaluate the influence of the operating parameters on the degree of abrasive tool wear, the lowest and highest values of the test results were summarized, and are given in Table 4. According to this table, the most influential operating parameter in the polishing process is the head rotation speed, because the degree of abrasive tool wear for this parameter has the highest value (0.866 g) at a variation of +50%. The corresponding values for head pressure and water flow rate are 0.853 and 0.745 g, respectively. Therefore, it can be said that in the polishing stage of building stones at processing plants, the most influential parameter, which has the most significant impact on the wear of the abrasive tools, is the head rotation speed or sliding distance.

**Table 4.** The highest and lowest values of abrasive tool wear in different levels of operating parameters.

| Operating Parameter | Abrasive Tool Wear (g) | | | | |
|---|---|---|---|---|---|
| | −50% | −25% | +0.25% | +50% | +75% |
| Head pressure | 0.109–0.520 | 0.186–0.632 | 0.276–0.849 | 0.238–0.853 | - |
| Head rotation speed | 0.065–0.523 | 0.206–0.652 | 0.374–0.796 | 0.460–0.886 | - |
| Water flow rate | 0.338–0.745 | 0.306–0.719 | 0.221–0.668 | 0.174–0.556 | 0.161–0.546 |

## 4. Conclusions

In the polishing stage of processing plants, operating parameters are crucial factors in the wear of abrasive tools. In this study, to investigate the effect of the operating parameters on the wear of abrasive tools, three main parameters—including polishing head pressure, head rotation speed and water flow rate—were considered. For better clarity and visibility of wear results, and to investigate the wear trends, the values of the operating parameters were changed within a range between ±25% and ±50% of the standard conditions. The abrasivity tests were carried out on fifteen different types of Iranian granitic building stone, and abrasive tool wear was determined for all samples. On the basis of the results, the following conclusions could be drawn:

- By increasing the polishing head pressure, a nonlinear behavior was observed in the wear of the abrasive tool, whereby it first increased, up until a critical pressure of around 6.25 bar, after which it gradually decreased. This nonlinear behavior is inconsistent with the well-known classical Archard's law. The FESEM micrographs of the worn pin surfaces showed that the wear mechanisms were not formed completely at pressures greater than 6.25 bar due to excessive loads, and the debonded abrasive particles could not be pulled out from the pin surface, i.e., interlocking between the pins and the stone surfaces had occurred. As a result, there was a decrease in the amount of abrasive tool wear.
- There was a positive linear relationship between the abrasive tool wear and the head rotation speed. In samples with low and moderate abrasivity, the wear was approximately proportional to the head rotation speed. However, this trend for samples with high abrasivity ($S_7$ and $S_{13}$) was less marked than for the other samples. This is may be due to the heterogeneity of the stones and pins in the subsurface layers, causing this behavior to be partly disproportional. The FESEM micrographs also showed that, with increasing the rotation speed, besides the abrasive and adhesive wear, delamination wear also resulted in many flakes on the pin surface. As a result, the loss of abrasive tools was increased.
- When the water flow rate was increased to 5 L/min, there was a slight decrease in the wear of the abrasive tool, after which it dropped suddenly for almost all samples. Then, the wear of the abrasive tool remained nearly unchanged at water flow rates greater than of 6 L/min.
- The results revealed that, among the investigated operating parameters, the head rotation speed had the most significant impact on the abrasive tool wear.
- Although the adhesion and delamination wear mechanisms were observed in the FESEM photomicrographs of worn surface of pins, the dominant wear mechanism due to variation of polishing operating parameters was the abrasive wear mechanism.

Finally, it should be pointed out that the main aim of this study was to examine the effect of operating parameters on the abrasive tool wear on selected granitic building stones. To check the general validity of the obtained results, further research should be carried out to address other building stone varieties. The present research results can be used in the processing plants for building stone, ceramic polishing, and factories for abrasive tool production.

**Author Contributions:** Conceptualization, A.F., E.G. and S.H.H.; methodology, E.G. and S.H.H.; software, A.F.; validation, A.F., E.G. and S.H.H.; formal analysis, A.F. and E.G.; investigation, A.F.; resources, A.F. and E.G.; data curation, A.F. and E.G.; writing—original draft preparation, A.F.; writing—review and editing, E.G.; visualization, E.G. and R.B.; supervision, E.G., S.H.H. and R.B.; project administration, E.G. and S.H.H. All authors have read and agreed to the published version of the manuscript.

**Funding:** This research received no external funding.

**Data Availability Statement:** Not applicable.

**Conflicts of Interest:** The authors declare no conflict of interest.

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
