# Peer review of "Investigating the Effect of Operating Parameters on the Wear of Abrasive Tools in the Polishing Stage of Granitic Building Stones"

_lubricants, doi:10.3390/lubricants10110321_

Round 1
Reviewer 1 Report
· (1) The introduction section should provide an intensive review of the previous works carried out to investigate the polishing of granite, and thus the research gap. However, this section only gives some very general information about polishing.
(2) Needs to explain the materials of the pins.
(3) Move table 2 and the explanation for this table to “Results and discussion” section.
(4) What does the ±25% and ±50% mean?
(5) Move table 3 and the description of the pressure, flow rate and speed used to section 2.2.
(6) Figure 9 in page 9 line 228 should be Figure 8.
(7) Descriptions of the wear mechanisms (especially on the delamination) seen in the FESEM images are not convincing.
(8) The main concern is the lack of novelty of the paper. It does not provide explanation on the different wear rate exhibited by the sample tested.
Author Response
Dear Reviewer
Thank you for your attention, valuable comments and guides. We modified manuscript based on your comments. The changes have been shown in highlighted font in the manuscript. If you think that manuscript needs more revision, please let us know. You can find the answers to comments in the following:
1) The introduction section should provide an intensive review of the previous works carried out to investigate the polishing of granite, and thus the research gap. However, this section only gives some very general information about polishing.
- Thanks for the comment. An intensive review of the previous works has been provided on page 3 (lines 81-103).
2) Needs to explain the materials of the pins.
- The materials of the abrasive tool (pin) have been explained and highlighted on page 6 (lines 159-164), based on the XRD results.
3) Move table 2 and the explanation for this table to “Results and discussion” section.
- Thanks for this useful comment. This shift has been done and highlighted in the manuscript on page 8 (line 220).
4) What does the ±25% and ±50% mean?
- As can be seen in table 2, the amounts of abrasive tool wear due to size of pins are very little (in terms of gram). For better clarity and visibility of wear results and to investigate wear trends due to variation of operating parameters, we decided to change the operating parameters between ±25% and ±50% of standard conditions. In other words, the variations would be -50, -25, +25 and +50 percent for any operating parameters. For example, in the case of head pressure in table 2, if we consider standard pressure of 5 bar, we will have:
5+5*(50%) =7.5 bar, 5+5*(25%) = 6.25 bar, 5-5*(0.25%) = 3.75 bar and 5-5*(50%) = 2.5 bar.
5) Move table 3 and the description of the pressure, flow rate and speed used to section 2.2.
- Thanks for this nice comment. This shift has been done in the manuscript on page 7 (line 204).
6) Figure 9 in page 9 line 228 should be Figure 8.
- The correction has been done in line 280.
7) Descriptions of the wear mechanisms (especially on the delamination) seen in the FESEM images are not convincing.
- The wear mechanisms have been described and highlighted on page 10 (lines 285-296).
8) The main concern is the lack of novelty of the paper. It does not provide explanation on the different wear rate exhibited by the sample tested.
- There are many previous studies on the polishing process of granitic building stone, but the investigations have been carried out mainly for increasing the surface appearance quality of the stones, such as glossiness; while our study has been focused on the effect of operating parameters on the abrasive tool wear and their mechanisms during the polishing process of granitic building stone. No paper has been published in this field so far, to the best of authors’ knowledge. These explanations are added and highlighted on page 2 (lines 66-77).

Reviewer 2 Report
In the manuscript ID lubricants-2015425, the authors studied the effects of polishing head pressure, head rotation speed, and water flow rate on the wear of abrasive tools in the polishing stage on the fifteen different types of Iranian granite building stones. The test's standard operating parameters (head pressure of 5 bar, head rotation speed of 300 rpm, and water flow rate of 4 l/min) were modified between ±25% and ±50%. The 630 experimental test was conducted as follows: (i) at 300 rpm Head rotation, Speed, and 4l / min Water flow rate they varied head pressure from 2.5 to 7.5 bar, (ii) at 300 rpm Head rotation Speed and 5 bar head pressure they varied Water flow rate from 2 l/min to 7 l/min, and (iii) at 5 bar head pressure and 4l/min Water flow rate they varied Head rotation speed from 150 rpm to 450 rpm. They also investigated the FESEM micrographs of the worn pin surfaces after performing the test on granodiorite stone (S2) with different pressures of (a) 2, (b) 3.75, (c) 6.25, and (d) 7.5 bar. After analyzing the experimental results and FESEM micrographs they concluded that (a) the wear mechanisms were not formed entirely due to excessive loads, and the debonded abrasive particles could not be pulled out from the pin surface, leading to the decrease of the abrasive tool wear; (b) by increasing the polishing head pressure the abrasive tool wear was increased up to a critical pressure of 6.25 bar; then it was gradually decreased, (c) abrasive tool wear was decreased with increasing the water flow rate up to 6 lit/min and at a rate higher than this one, was almost unchanged.
The work, as a whole, is of interest to researchers of Materials Science and Engineering and especially, in Construction and Building Materials, and can be developed in the processing plants of building stone after more investigations and further research.
As a result, I agree to be published in this form, with the correction of minor spelling errors that appear in the manuscript, such for example:
Please verify all the symbol units and modify them according to SI units. For instance, in Line 15 and tables, SI units symbol of Liter „l” or „L”, in „gr” is „g” etc.
Line 57: Please correct the SI units in the correct one ( m3)
Line 149 Put the degree of exponent: o C
Line 229: specifications about Figure 9 are before Figure 8-a and Figure 8-b
Lines and 178, 250, 274 specify the number of the figure.
Author Response
Dear Reviewer
Thank you for your attention, valuable comments and guides. We modified manuscript based on your comments. The changes have been shown in highlighted font in the manuscript. If you think that manuscript needs more revision, please let us know. You can find the answers to comments in the following:
1) Please verify all the symbol units and modify them according to SI units. For instance, in Line 15 and tables, SI units symbol of Liter „l” or „L”, in „gr” is „g” etc.
- The corrections have been done through the all text.
2) Line 57: Please correct the SI units in the correct one (m3)
- The correction has been done on page 2 (line 57).
3) Line 149 Put the degree of exponent: o C
- The correction has been done in page 8 (line 210).
4) Line 229: specifications about Figure 9 are before Figure 8-a and Figure 8-b
- The correction has been done on page 10 (line 280).
5) Lines and 178, 250, 274 specify the number of the figure.
- The correction has been done in lines 228, 306 and 330.

Reviewer 3 Report
This paper reports about the wear of a counterpart in polishing different stopes under three different parameters: contact pressure, test speed, water flow. The paper is clear and well structured. I miss some more analysis of the data, which I will explain below. Therefore, I would recommend the following minor revision:
1. Abstract and text: the authors mention a ‘critical value’ of load of 6.25 bar. The authors should ‘soften’ that claim, since they do not have large resolution about that value. In other words, the critical load could be 6.0, 6.20, 6.10, 6.30 bar, etc. (strictly speaking, any value between 5 and 7.5 bar). Therefore, please refer to values ‘around 6.25 bar’.
2. Figure 2: please add the scale bar of the images, or indicate the field of view of the images.
3. The experimental details need to be clarified further:
a. Line 101: please indicate the preparation for the thin sections of the petrographic characteristics.
b. Please indicate the conditions of the FESEM (equipment, voltage, etc.).
c. In the text, XRD measurements are reported (line 199). Please indicate the equipment and approach employed.
d. Indicate the load employed on the tests (i.e. not only the pressures), and also the dimensions of the pins.
e. Line 309: please indicate the ‘thermometer device’ employed in that measurement.
4. Table 1: indicate the units of the %.
5. Line 147: the results of the Table 2 should be included in the Section of ‘Results and Discussion’.
6. In general, I believe that the authors could get more information of the results presented. They get ‘overall behaviors’, but they spend little time to discuss the differences between different stones. In other words, the different stones studied ‘just’ allow confirming a ‘common behavior’. Therefore, I would recommend the authors to perform the following analyses, in order to enhance the interest of the paper:
a. Since several measurements have been carried out, I miss the presence of error bars.
b. The mass loss can be transformed into ‘variation of pin height’, just by using the original mass and dimensions of the pins. This would allow to get an easier understanding of the meaning of the ‘mass loss’. In addition, such calculation could help on the ‘surface and subsurface’ discussions.
c. I would strongly recommend the authors to perform linear fittings in Figures 6 (up to 6.25 bar) and Figure 9. The slopes and ordinate at the origin would help to distinguish the behaviors of different stones, and correlate them with the properties of the stone (e.g hardness).
d. Line 193: in line with the previous comment, please add the values of hardness of the different stones; it would be very interesting to correlate the hardness of stones against the wear of the pins: are harder stones wearing the pins more?
e. Figure 11: my eyes may be wrong, but I see a kind of ‘step’ in the behavior of the wear rate with the water flow. For instance, in S7, I see a more or less constant value of wear loss until 5 l/min, where a sudden drop is observed, and from 6 l/min the value is more or less constant again. The location and ‘size’ of that step seems to be stone dependent (S12 from 4 to 6, S15 from 4 to 5…), but if it is a ‘consistent behavior’, it should be commented in the text.
7. Line 199: please indicate the matrix where the SiC particles are embedded, and the hardness of that matrix.
8. Figure 12: I fail to understand what is observed in the figure. What is the orange circle in the inset? Does it correspond with the wear track? Please add legends for the meaning of the color, and a scale bar for the dimensions.
9. Lines 340-342: please elaborate that paragraph, with clear explanations of the values observed. In terms of rotation speed, does the wear merely scales with the increased length of the tests, or is there any additional effect? In other words, double the speed ® double the length ® double the wear? Please, comment.
10. In connection with the previous comment, the authors must be very careful with the ‘influence of the rotation speed’; the reason is that, together with that parameter, they are changing the test length. Therefore, it is unclear the influence of each parameter. Please, comment.
11. Minor comments:
a. Line 31: polishing operation, in singular.
b. Line 57: write the m^3 without the ‘^’ and the 3 in superscript.
c. Line 57: add the units of the wear coefficient (non-dimensional?).
d. Line 149: degree Celsius as superscript.
e. Line 192: stones, not sones.
f. Line 192: replace ‘several times more’ by ‘several times higher’.
g. Please, check if the symbol for ‘liters’ is ‘L’ and not ‘l’. If it is the case, please correct through the paper.
Author Response
Dear Reviewer
Thank you for your attention, valuable comments and guides. We modified manuscript based on your comments. The changes have been shown in highlighted font in the manuscript. If you think that manuscript needs more revision, please let us know. You can find the answers to comments in the following:
1) Abstract and text: the authors mention a ‘critical value’ of load of 6.25 bar. The authors should ‘soften’ that claim, since they do not have large resolution about that value. In other words, the critical load could be 6.0, 6.20, 6.10, 6.30 bar, etc. (strictly speaking, any value between 5 and 7.5 bar). Therefore, please refer to values ‘around 6.25 bar’.
- Exactly, you are right. Thanks for your precise comment. Both the abstract and text have been corrected and highlighted on lines 19, 229, 302,424.
2) Figure 2: please add the scale bar of the images, or indicate the field of view of the images.
- The scale bar was added to this figure (page 4).
3) The experimental details need to be clarified further:
- Line 101: please indicate the preparation for the thin sections of the petrographic characteristics
- In page 3 (lines 131-139), the preparation for the thin sections has completely been described and highlighted.
- Please indicate the conditions of the FESEM (equipment, voltage, etc.).
- Type of equipment and its operational condition have been indicated on page 10 (lines 278-280).
- In the text, XRD measurements are reported (line 199). Please indicate the equipment and approach employed.
- The equipment and used approach have been indicated and highlighted on page of 6 (lines 159-164).
- Indicate the load employed on the tests (i.e. not only the pressures), and also the dimensions of the pins.
- According to the dimension of pins used in the test, the sum areas of two pins equal one square centimeter. As a result, the amount of pressure and load is the same. This sentence has been added and highlighted on page 6 (lines177-179).
In addition, the dimensions of the pin have been shown in Figure 4 on page 6.
- Line 309: please indicate the ‘thermometer device’ employed in that measurement.
- Specification of thermometer used in the test are indicated and highlighted on page 14 (lines 375).
4) Table 1: indicate the units of the %.
- As can be seen at the top of the table 1, the phrase “Modal composition (%)” has been written.
5) Line 147: the results of the Table 2 should be included in the Section of ‘Results and Discussion’.
- This Table was added to the section results and discussion based on your comment (page 8).
- In general, I believe that the authors could get more information of the results presented. They get ‘overall behaviors’, but they spend little time to discuss the differences between different stones. In other words, the different stones studied ‘just’ allow confirming a ‘common behavior’. Therefore, I would recommend the authors to perform the following analyses, in order to enhance the interest of the paper:
- Since several measurements have been carried out, I miss the presence of error bars.
- As mentioned in the text (section 2.2), the test has been repeated at least three times and average of three measurements has been reported. Unfortunately, due to the large number of results, it was impossible to present them in a table in the text and the average results have inevitably been shown in figures 6, 9 and 11. Moreover, as can be seen in these figures, they will not be clear by inserting error bars because the lines of the graphs very close to each other. We are completely ready to send the table of results if you want.
- The mass loss can be transformed into ‘variation of pin height’, just by using the original mass and dimensions of the pins. This would allow to get an easier understanding of the meaning of the ‘mass loss’. In addition, such calculation could help on the ‘surface and subsurface’ discussions.
- To reply this comment, it should be said that in most of wear tests in field of rock mechanics such as Taber abrasion test, the NTNU (Norwegian University of Science and Technology) tests (AV and AVS), LCPC (Labratorie Central des ponts et chaussees), Los Angles abrasion test (LA), Rolling Indentation Abrasion Test (RIAT), Schimazek’s pin on disk test, weight loss of abrasive tool was considered as wear index. Accordingly, in our new test, weight loss of the abrasive tool (pins) has been measured and developed as building stone abrasivity index (BSAI). You can see the detail of the new test in our previous paper [20]. For observing the paper, you can refer to following link:
https://doi.org/10.1016/j.conbuildmat.2021.124497
Of course, measuring of variation of pin height will be a great idea for our future research, and we will investigate it.
- I would strongly recommend the authors to perform linear fittings in Figures 6 (up to 6.25 bar) and Figure 9. The slopes and ordinate at the origin would help to distinguish the behaviors of different stones, and correlate them with the properties of the stone (e.g hardness).
- In this study, our main aim is to investigate the effect of operating parameters on the wear of abrasive tools in the polishing process of building stone. The relationship between tool wear (pins) and physico-mechanical properties of the samples such as hardness has been examined in our previous paper [20].
- Line 193: in line with the previous comment, please add the values of hardness of the different stones; it would be very interesting to correlate the hardness of stones against the wear of the pins: are harder stones wearing the pins more?
- Yes, exactly, the harder stone is, the greater pin wear is. We have carried out and reported this correlation in our previous paper [20]. We have shown that there is strong correlation between the pins wear and Mohs hardness.
- Figure 11: my eyes may be wrong, but I see a kind of ‘step’ in the behavior of the wear rate with the water flow. For instance, in S7, I see a more or less constant value of wear loss until 5 L/min, where a sudden drop is observed, and from 6 L/min the value is more or less constant again. The location and ‘size’ of that step seems to be stone dependent (S12 from 4 to 6, S15 from 4 to 5…), but if it is a ‘consistent behavior’, it should be commented in the text.
- Thank you for your scrutiny and nice comment. This behavior has been described and highlighted on page 13 (lines 361-365).
- Line 199: please indicate the matrix where the SiC particles are embedded, and the hardness of that matrix.
- Specifications of the material of the pin have been indicated and highlighted on page 6 (lines 159-164).
- Figure 12: I fail to understand what is observed in the figure. What is the orange circle in the inset? Does it correspond with the wear track? Please add legends for the meaning of the color, and a scale bar for the dimensions.
- The orange circle shows the wear track that has the most temperature (just over 52 °C). The legends and scales have been inserted in figure 12 on page 14.
- Lines 340-342: please elaborate that paragraph, with clear explanations of the values observed. In terms of rotation speed, does the wear merely scales with the increased length of the tests, or is there any additional effect? In other words, double the speed ® double the length ® double the wear? Please, comment.
- You have a good point there. As shown in Figure 9, by increasing the head rotation speed (sliding distance) abrasive tool wear has enhance. However, the abrasive tool wear is approximately proportional to head rotation speed i.e. if the head rotation speed is doubled (length two times), the abrasive tool wear won’t become twice precisely. One of the reasons may be the heterogeneity of stones and pins in the subsurface layers, which causes this behavior to be partly disproportional.
- In connection with the previous comment, the authors must be very careful with the ‘influence of the rotation speed’; the reason is that, together with that parameter, they are changing the test length. Therefore, it is unclear the influence of each parameter. Please, comment.
- As mentioned in line 183, the testing time duration is five minutes. However, following your useful comment and to check the effect of the distance travelled by the pins, we carried out further tests at the standard operating conditions (head pressure of 5 bar, head rotation speed of 300 rpm and water flow rate of 4 L/min) for time durations of 10, 15, and 20 minutes. The test accomplished for all samples and results of samples 1, 5, 7, and 15 have been shown in the following figures:
As can be seen in the above figures, the abrasive tool wear is partly proportional to testing time or sliding distance.
- Minor comments:
- Line 31: polishing operation, in singular.
- The correction has been done in line 31.
- Line 57: write the m^3 without the ‘^’ and the 3 in superscript.
- The correction has been done in line 57.
- Line 57: add the units of the wear coefficient (non-dimensional?).
- Yes, this coefficient is non dimensional.
- Line 149: degree Celsius as superscript.
- The correction has been done in page 8 (line 210).
- Line 192: stones, not sones.
- The correction has been done in page 9 (line 242).
- Line 192: replace ‘several times more’ by ‘several times higher’.
- The correction has been done in page 9 (line 242).
- Please, check if the symbol for ‘liters’ is ‘L’ and not ‘l’. If it is the case, please correct through the paper.
- The corrections have been done through the all text.

Round 2
Reviewer 1 Report
The authors have addressed all the questions, and the quality and readability of the revised paper have been improved. I recommend the paper to be accepted for publication.